# Exploring the Impact of an Innovative Peer Role-Play Simulation to Cultivate Student Pharmacists’ Motivational Interviewing Skills

**DOI:** 10.3390/pharmacy11040122

**Published:** 2023-07-29

**Authors:** Paul Denvir, Laurie L. Briceland

**Affiliations:** 1Department of Population Health Sciences, Albany College of Pharmacy and Health Sciences, Albany, NY 12208, USA; paul.denvir@acphs.edu; 2Department of Pharmacy Practice, Albany College of Pharmacy and Health Sciences, Albany, NY 12208, USA

**Keywords:** patient-centered communication, pharmacist–patient communication, empathy, motivational interviewing, peer role-playing simulations, professional identity formation

## Abstract

Effective patient-centered communication is a foundational skill for student pharmacists, with recent decades broadening the scope of professional responsibilities to include an increased emphasis on empathic communication and motivational interviewing (MI) as tools to support patients’ therapeutic adherence. Role-play is a potentially effective pedagogical approach to cultivate these skills, although previous research has identified common shortcomings that can hinder its educational value, particularly in peer role-play with relatively inexperienced learners. The purpose of this study is to describe and provide initial assessment data for an innovative approach to peer role-play that incorporates pedagogical principles to address these common shortcomings. Using a mixed-methods study design that includes instructor-graded rubrics and inductive thematic analysis of student reflections, our findings indicate that students successfully demonstrated a range of important competencies through this experience and perceived it to be both challenging and highly beneficial for their personal and professional development. Among the MI principles and techniques practiced, students performed especially well on expressing empathy and frequently reflected on its importance for future patient care and clinical collaborations. Our findings also suggest that peer engagement through team activities and partnered role-play provides a felicitous context to explore empathic communication together.

## 1. Introduction

Learning how to effectively provide patient-centered communication using the Pharmacists’ Patient Care Process [1] is a foundational skill for student pharmacists [2]. The pharmacy profession has a long history of counseling patients about safe and effective medication use; many patients struggle to afford and/or adhere to complex medication regimens for chronic illnesses and with behavioral health challenges like tobacco cessation [3]. Recent decades have seen an expansion in the scope of responsibilities for effective pharmacist–patient communication, including reflective listening with empathic responding, the use of open-ended questioning, and motivational interviewing techniques [4,5]. While the pharmacist’s role as the medication specialist is essential [6], the practical value of that expertise could be limited if effective patient communication skills are not adequately developed, as studies have demonstrated improved patient outcomes when provider– and pharmacist–patient communication is optimized [7,8,9,10]. Thus, it is imperative that student pharmacists are introduced to effective pharmacist–patient communication methods in the first professional year (P1), providing ample time to develop and receive feedback on these essential skills throughout the duration of the curriculum [5,10,11].

### 1.1. Motivational Interviewing in Pharmacy

Motivational interviewing (MI), originally developed in large part by Miller and Rollnick in the 1980s [12], is an evidence-based approach to patient/client counseling that has emerged as an effective model for a wide variety of behavioral health concerns [13]. MI has received considerable empirical support and broad adoption within the field of pharmacy [12,14]. MI is a person-centered counseling approach that guides patients to articulate and explore their ambivalence about change and enhances intrinsic motivation on the patient’s own terms and in their own voice. For healthcare providers, MI emphasizes empathic communication through reflective listening and respect for patient autonomy by avoiding confrontational or paternalistic counseling responses. Rather, the core focus of MI is to guide patients to engage in “change talk,” which refers broadly to any talk that acknowledges the downsides of the status quo (not changing) or optimism and self-efficacy about change. Pharmacists and pharmacy educators have adopted MI as a foundation for patient-centered communication, particularly in terms of improving medication adherence [14,15,16]. A recent study examining P1 student MI perceptions and attitudes determined that confidence in counseling skills was a predictor that students would incorporate MI and other patient-centered communication skills into eventual practice. Those authors concluded that there remains a need for colleges to incorporate innovative exercises in patient-centered communication, including MI, into their curricula, so that students can develop the necessary confidence to carry these important skills into practice [17]. We have responded to this challenge by designing an engaging set of exercises aimed at cultivating P1 pharmacy students’ MI skills and confidence in patient-centered communication.

### 1.2. Simulations and Role-Play Teaching and Learning in Pharmacy Education

One pedagogical genre of particular interest in pharmacy education is training simulation, an active-learning instructional technique used to develop knowledge, skills, and attitudes with the goal of transferring the learning into actual practice [16,18,19]. Simulations mimic authentic experiences by employing critical aspects of the clinical scenario, enabling the learner to practice in a safe environment and gain confidence, without risk of harm to an actual patient [5]. A recent publication regarding P3 student pharmacists determined that a combination of didactic and skills-based simulation training resulted in an improvement in empathy scores; these students indicated that the opportunity to practice empathy skills (a major component of MI) in the simulated environment proved to be a valuable experience [20]. The MI innovation described herein includes the combination of didactic instruction followed by simulation exercises, one specifically involving role-play. Role-play is one type of simulation, “an artificial representation of a real-world situation to achieve educational goals through experiential learning” [21]. Role-play exercises are used in myriad settings, including educational contexts in which professionals need to cultivate specific communication skills. Precepts of clinical education and patient-centered care, focusing on empathic communication, respect for patient agency/autonomy, and shared decision-making models can be practiced in well-designed role-play [5,10]. In role-play, typically the learner plays the role of the health professional, and the simulated patient role is played by a “standardized patient”, who could be a paid or volunteer actor, lay person, real patient [21], or faculty member [15]. One of the core challenges of effective role-play centers on authenticity (or realism) [22]. For the benefit of learners, role-play is often idealized or oversimplified—things tend to go according to plan [23]. In high-stakes fields like health and medicine, it is understandably necessary to provide safe spaces to cultivate skills without the risk of harming others [5], yet it is also fair to say that this is not fully authentic to real-world practice. Scholars have developed a variety of innovative approaches to realism in role-play, such as the conversation analytic role-play method (CARM) which uses audiovisual recordings and transcripts of real-world encounters to stimulate reflection and discussion of communication choices as learners vicariously “live through” an encounter moment by moment [22]. Our innovation incorporates aspects of CARM, such as its close attention to turn-by-turn sequences of spoken interaction.

The use of a standardized patient (SP) as an active-learning role-play strategy in teaching MI skills to P1 pharmacy students has been shown to be the most effective teaching modality when compared to written dialogue and peer role-plays [10]. However, using SPs in the curriculum, especially if multiple practice opportunities or different activities are included for a large number of learners, can prove to be labor-intensive and prohibitively costly [10]. Peer role-play, in which two learners are paired and in turn each play the role of the patient and healthcare provider, has emerged as an engaging and less costly alternative to using the SP, with some research demonstrating similar effectiveness in developing MI skills when comparing peer role-plays to SP role-plays [24]. Moreover, peer interactions offer additional benefits by ensuring active engagement and increasing student learning by participating in partner’s scenarios, as well as inculcating feelings of confidence, comfort, and inclusivity within a didactic course [25,26,27]. Given these collective benefits, we elected to utilize a well-designed peer role-play simulation for the innovation described herein.

Peer role-play can present unique authenticity challenges, especially in terms of naturalistic organization of social interaction. As most educators (including ourselves) who have used peer role-play can attest, such exercises tend to go wrong in one of the following ways: (i) the experience is overly scripted and unrealistic, so learners end up mechanically “reading” scripts at one another and do not practice authentic listening, engagement, or responsiveness [23,24]; (ii) the experience is overly open, unstructured, or undefined, so learners do not know what they are supposed to do or say, how to move the interaction forward, or what their goals are [5,21]; (iii) the learner playing the role of the healthcare professional may not have sufficient clinical knowledge of the patient scenario and may provide misinformation during the encounter, confounding the original intent of the exercise which was to practice communication skills [15]; and (iv) as peer learners, both parties are relatively inexperienced and may struggle to present a reasonable and useful challenge for one another. In many cases, peer learners may err on the side of cooperation, helping one another demonstrate the skills that the assignment is designed to cultivate [24]; this amiable approach may help to build trust and confidence, but it could also leave students unprepared for the unanticipated difficulties and resistance they will encounter with real-world interactants. We have turned these peer role-play challenges into opportunities for improvement in our MI exercise. We describe herein a peer role-play approach using innovative strategies to address these known shortcomings of traditional role-play exercises by balancing the need for both structure and improvisation in a realistic format. The aim of our study was to assess the impact of our peer role-play simulations on cultivating P1 student pharmacists’ motivational interviewing skills.

## 2. Method

This MI peer role-play project was conducted at Albany College of Pharmacy and Health Sciences (ACPHS), a private college in New York which offers a traditional four-year Doctor of Pharmacy program. The project underwent an Institutional Review Board review and met the criteria for exemption from the requirements of federal regulations.

### 2.1. Course Context and Preparatory Activities for Role-Playing

During the Fall 2022 semester, 82 P1 students in the required two-credit Foundations of Pharmacy (FOP) course completed a three-week scaffolded series of MI learning activities. These learning activities included required reading, didactic instruction, and an in-class small peer group interaction, with subsequent outside-of-class group interaction to complete an end-of-week assignment upload. The topic of Week 1 was “Medication Adherence/Health Literac” the topic for Week 2 was “Setting the Stage: Empathic Communication and Motivational Interviewing (MI) Strategies in Pharmacy Practice”. For Week 2, students completed four MI formulation challenges created by the course instructors (authors), in which teams of four analyzed scripts of fictional exchanges between a pharmacist and a patient, one for each of four key MI counseling strategies (i.e., expressing empathy, developing discrepancy, rolling with resistance, and supporting self-efficacy) [12]. These formulation challenges were designed to highlight the importance of word choice and the need to tailor MI strategies to specific details and moments of interaction with patients, and these were intended to serve as practice for the final Week 3 MI learning activity. See Appendix A for more detail on the formulation challenges. The topic for Week 3 was “MI Peer Role-Playing to Improve Medication Adherence”. Student teams of four informally practiced their skills during class time, using the four MI techniques introduced the week prior and the role-play activity provided. The students received informal verbal feedback from team members and instructors. After class, the student teams were subdivided into pairs; each pair prepared formal video recordings demonstrating MI skills. This learning activity, the focal topic of this paper, is described in detail in Section 2.2, below. The learning objectives for the MI formulation challenges and peer role-playing exercises were mapped to 10 different Center for the Advancement of Pharmacy Education (CAPE) educational outcomes [28] and are shown in Table 1.

### 2.2. Description of Focal Learning Activity: An Innovative Approach to Peer Role-Play in MI

The Fall 2022 FOP offering of the MI peer role-play simulation was our third iteration of this assignment, with P1 class sizes of 127, 165, and 82 students, respectively. Given these relatively large class sizes, we opted to employ an online platform (versus in-person) for the peer role-play activity, which proved especially effective given that the COVID-19 pandemic was in full swing during this period. Our decision to employ videoconferencing was driven primarily by its efficiency and flexibility for students to record when their schedules aligned. Each pair of students was required to coordinate, schedule, and record two MI counselling videoconferences using the Zoom platform (Santa Clara, CA, USA), with partners taking turns playing the roles of student pharmacist and patient; for each pair, two videos were submitted, demonstrating each student playing the roles of pharmacist and patient. The intended duration of the MI interview was 6–8 min, although students who were outside of this range were not penalized. Partners were free to record as many counseling sessions as desired, submitting their best version for a grade.

### 2.3. Learning Materials and Activities

In previous iterations of this assignment, students often inadvertently misrepresented aspects of the simulated clinical scenario. As P1 students, many lacked the pharmacotherapeutic background to speak accurately about medications and associated disease states, yet they were understandably preoccupied with that aspect of the counseling process. To address this issue, we developed a unique innovation, a disease-specific primer.

#### 2.3.1. Asthma and Medication Primer

A streamlined “Asthma and Medication Primer” document (Document S1: Asthma and Medication Primer in Appendix A) was developed for teaching purposes. Asthma was selected because it is a prevalent disease state in many populations, is routinely treated with pharmacotherapy, and is a recurrent area of pharmacist–patient adherence counseling [3]. The primer provides accurate but simplified information about asthma as a disease state, as well as two fictional asthma medications: Alpha, a maintenance inhaler, and Omega, a rescue inhaler. Alpha and Omega are based on actual asthma therapies with the purpose of simulating a common dynamic in asthma adherence counseling, which is navigating a patient’s overreliance on rescue inhalers rather than maintenance inhalers [29]. Students were instructed to use only the information in the primer during their counseling sessions, putting the focus instead on patient-centered communication and MI, rather than the nuances of pharmacotherapy. Our use of a reality-based fictional primer was one innovative method by which we balanced the need for authenticity (a genuine behavioral health challenge rooted in medication adherence) and the need to simplify and narrow the focus on communication skill development.

#### 2.3.2. Role-Play Scenario and Roles

The pharmacist–patient simulation takes place in an asthma clinic associated with a large urban hospital system. Patients are referred to the asthma clinic if they present in the emergency room with asthma symptoms or if they are struggling to adhere to their asthma regimens. The clinic has a built-in pharmacy and is staffed primarily by pharmacy preceptors and students on advanced pharmacy practice experiences (APPE), all of whom are trained to adopt MI principles in their adherence counseling. In this role-play, the learner playing the healthcare provider takes on the role of an APPE student at the clinic, functioning as a student pharmacist, while the partnered student plays a patient. The instructors designed this scenario with realism in mind, as students are likely to experience the (student) pharmacist’s role in asthma education during future practice opportunities. For guidance in assuming the student pharmacist and patient roles, participants were provided brief biographies, to be expanded through improvisation, as well as a goal to bear in mind during the interaction, as described in Table 2.

While we expected students to appreciate the educational value of practicing MI principles in the role of a pharmacist, we also stressed that there is important professional development to be gained from embodying the patient role. Research suggests that portraying a character in a role-play scenario, especially if that portrayal encourages improvisation and personalization, is an opportunity to cultivate empathy for the challenging circumstances that people navigate in their worlds [21]. Giving voice to those challenges not only provides opportunities for the “pharmacist” to practice empathic responses but also for the “patient” to imagine and inhabit circumstances they may not have encountered in their own lives. Patient-centered care depends upon a provider’s ability to take their patients’ perspectives [30]; thus, we theorized that experiencing both sides of this encounter would be especially valuable. To emphasize the importance of the patient role in the role-play exercises, we revamped the grading rubric to assign points for how well the learner played the role of the patient (see Section 2.4 for grading rubric description); by doing so, we also addressed the shortcoming that the peer playing the patient was too amiable and did not provide the role-playing pharmacist an adequate challenge.

#### 2.3.3. Turn-by-Turn Action Script

To address a shortcoming identified in our earlier iterations of the assignment in which students created overly scripted text for the role-play, the instructors wrote the script for the MI activity described herein. The revamped script also standardized the experience for all students by incorporating all four MI strategies, affording each student a broader and fuller experience of MI. To guide students on how to engage in the role-play activity turn-by-turn, students were provided with a PowerPoint slide set containing detailed information about the patient care scenario (Section 2.3.2 above) and a “Turn-by-Turn Action Script”, another key innovation in this role-play activity (See Appendix A: Turn-by-Turn Action Script). Figure 1 depicts the Zoom interface that students were instructed to use to record their counseling session; it includes both role-play participants’ video windows and a third window that contains the action script that guided the experience.

This interface allowed both students to see and hear one another while maintaining a mutual orientation to the ongoing requirements of the displayed action script. By performing and recording this way, instructors were also able to accurately assess student performances and their fulfillment of the required activities as the action script progressed. The action script employs an accessible visual framework in PowerPoint. On each slide, the upper portion depicts what the “patient” should do, and the lower portion depicts what the “student pharmacist” should do (i.e., their social actions). Each required action is shown in simple, color-coded smart art, serving as an ongoing visual reminder for both participants of what each should be doing to move the interaction forward. As the session progresses and the required actions are performed, the student role-playing as the “student pharmacist” is responsible for advancing the slides one at a time, helping to ensure that both participants are literally “on the same page” as they interact.

Increasing the degree of structure helped to address multiple concerns that novice role-players often experience in moving the interaction forward with purpose. In addition, the structure ensured that specific principles and skills of empathic communication and MI would reliably emerge during each counseling session. Creating a context for effective improvisation helped to address other concerns, as active listening and responsiveness is foundational for clinical practice. In previous versions of this assignment, many students simply took turns reading pre-prepared scripts; to help students gain confidence with thinking and speaking on their feet, we designed the action script to provide a limited and accessible context for improvisation. Students were never told exactly what they should say; they were given a basic social action and had to improvise the specific action formulation. Second, we designated certain actions in the script with the label “player’s choice”. For these actions, students were provided with two options for how to achieve the action, and they could select only one of the options. While students were free to engage in pure fiction, the assignment instructions encouraged them to borrow from their own life circumstances to the extent they felt comfortable. Importantly, students were instructed not to tell one another in advance which option they would select in any player’s choice. Rather, the “student pharmacist” would have to be ready to improvise their response based on their partner’s choice. While the player’s choice required a certain amount of improvisation for both parties, it was designed to create a relatively circumscribed set of possibilities, allowing the “student pharmacist” to reflect in advance about how they might respond to each trajectory without feeling overwhelmed by unpredictability. Figure 2 shows an example of the player’s choice in the action script.

### 2.4. Assessing the Impact of the Role-Play Activity

A mixed-methods design combining quantitative and qualitative data was employed in the analysis of the MI role-play activity. Two forms of data were used to assess the impact of this activity: (i) numerical results of instructor-graded rubrics, and (ii) written student reflections. In addition, we analyzed the video recordings and transcripts of the 10 highest-scoring and 10 lowest-scoring students to identify illustrative exemplars of stronger and weaker formulations that students produced while role-playing. These data complement the quantitative data gathered through the rubric by showing the accurate and often inventive choices that students could make within this innovative peer role-play, as well as areas where they may need additional support for future iterations of the assignment.

#### 2.4.1. Instructor-Graded Rubric

A 10-point grading rubric (out of 100 points in the course) was created by the instructors (authors) and used to guide the assessment and grading of the MI role-play video uploads. Students were graded on both their role-play as the student pharmacist (covering four MI strategies, for 8 points), and as the patient (covering preparation for the role and challenge to the pharmacist, for 2 points). For each MI strategy assessed, the rubric stated where in the action script slides the strategy should be displayed. One instructor graded all 82 videos using the rubric and provided detailed written feedback to each student, including an itemization of where within the video presentation the competency was not met. Rubric grades were collated from the gradebook in the learning management system and are reported in our results; the reasons why students lost points on the rubric are provided. The rubric is provided as Appendix A.

#### 2.4.2. Written Student Reflections

To gain a more nuanced first-hand perspective on this learning experience, a sub-set of students’ final course reflection papers were analyzed. The reflection paper was a required assignment in which students were free to select any course component from a provided list and respond to the following prompt: *Select one aspect of the course that made the greatest, most meaningful impact upon you/your learning, either positively or negatively. As a result of reflecting on this component, how will you specifically transform/change your behavior in the (immediate or longer term) future as you continue your journey as a student pharmacist, and eventually, pharmacist?*

To analyze the final reflection papers, we employed an inductive thematic analysis, a qualitative method well-suited to finding patterns in textual data [31]. Induction is an iterative, “bottom-up” analytic process in which generalizations emerge out of repeated exposure to data, rather than through the application of a priori theoretical concepts or categories. Our analysis proceeded in three waves, each supported by the qualitative analysis software Atlas.ti (Corvallis, OR, USA). In the first wave, the authors read each reflection and entered free-text comments to identify recurring topics and themes of interest, also known as open coding. Using insights from the first wave, the authors developed an initial coding framework with 40 categories, essentially a list of topics/themes that could be applied systematically to the full data set. In the second wave, the first author identified relevant text extracts in each reflection and assigned all applicable coding categories to each extract. Selecting the boundaries of a text extract requires some analytic judgment, but the aim is to capture the expression of a complete thought and its surrounding context. The second wave yielded 117 total extracts (5.9 per reflection) with an approximate length of two to four sentences each. The second wave was used to refine the initial codebook, eliminating codes that did not have adequate empirical support, combining codes that were revealed to share overlapping conceptual territory, and redefining code names to better convey the content and patterns in the data. The revised and final coding framework included 26 coding categories. In the third wave, the authors closely analyzed and discussed the interpretations of the 24 coded collections that were most relevant to the current analysis. This process yielded five main themes, as shown in the Results section.

## 3. Results

### 3.1. Instructor-Graded Rubrics

The MI role-play video assignment was completed by all 82 students. The average score on the grading rubric was 9.4 (out of 10.0) on the assignment, which demonstrates that the majority of students successfully demonstrated competency in each of the seven rubric criteria evaluated. Table 3 identifies the rubric criteria in which students lost partial or full points, indicating that the intended competency of that rubric criteria was not fully met. The two MI areas in which the most students lost points on the rubric were “develops discrepancy” and “patient autonomy”, in which 23 and 19 students, respectively, lost full or partial credit on that element. One student lost credit on the “expressing empathy” domain, with the 81 remaining students receiving full credit on the empathy domain.

Table 4 includes examples of quotes from student videos in which the role-playing student pharmacist demonstrated (or did not demonstrate) competency in each of the MI domains evaluated.

### 3.2. Written Student Reflections

Of the 82 students who submitted final course reflections, 20 (24%) chose to reflect on the MI experience, all of which were included in the analysis. The student commentary was overwhelmingly positive, highlighting the impact of close attention to communication options and choices, as well as the broad applicability of their learning to a range of personal and professional development domains. Many students commented on the realism of the scenario and reported feeling better prepared for encounters like this in their short- and long-term futures. They perceived the peer role-play as immersive and, in some cases, quite challenging but also reflected positively about the role of peer engagement in building both empathic awareness and confidence in their skill development. Table 5 presents the five key analytic themes, the raw number and percentage of extracts that spoke to each theme, and illustrative sample quotations.

## 4. Discussion

### 4.1. Unique Features of Our MI Approach That Improved Processes and Skills Assessment

Laying the foundation for the development of effective patient–provider communication skills within the first professional year is essential in pharmacy education [10,11,17]. Ekong has suggested that innovative exercises in patient-centered communication, including MI, be incorporated early into curricula, so that students can develop the confidence necessary to more predictably transfer these crucial communication skills into practice [17]. Our innovative MI exercises within a P1 required course incorporating formulation challenges and peer role-play provide a successful example of immersing first-semester P1 students into simulations to hone these important patient-centered communication principles and techniques. Our data corroborate findings that combining didactic exercises with practicing these skills is an effective methodology for student pharmacists to demonstrate competency in empathic communication techniques [20]; we too used a similar didactic/skills combination design and report exemplary results in the peer role-play (Table 3 and Table 4). We were pleased to find that only one student lost points for the inability to competently express empathy, as empathic communication was a large component of our didactic instruction in these exercises. The data in Table 3 also inform us of where to renew our efforts in future iterations of this assignment; for instance, under the category of “develops discrepancy”, where our largest number of students demonstrated struggles.

We attribute the positive findings noted herein to the many unique enhancements that we have implemented to address the previously mentioned shortcomings in earlier iterations. Notably, we have created an “Asthma and Medication Primer” (Appendix A) to address the issue of P1 students lacking sufficient clinical knowledge to accurately address pharmacotherapy nuances [15]. The primer enabled students to stay “on-task” and focus on the MI communication skills that they were honing, rather than discussing the pros/cons of specific medications (which students did not yet know). This primer concept can be readily adapted to other disease states, courses, and other Colleges of Pharmacy. Our second noteworthy innovation was the creation of the action script (Appendix A) which provided a turn-by-turn roadmap for the peer role-play. The action script addresses the shortcomings of overly scripted student-created role-plays which effectively preclude active listening and improvisation when playing the role of the pharmacist [23,24], coupled with providing a much-needed structure to effectively move the dialogue forward [5,21]. We also found that our students performed well when playing the role of the patient (Table 3). When compared to previous iterations of our exercise in which the action script was not available, this iteration yielded better-prepared role-playing patients who offered a balance of resistance and change talk, which is sometimes lacking in role-play [24]. The following is an example of one of our role-playing patient’s speaking turns, which exemplifies this important balance: *“You do bring up a good point on the benefits of Alpha reducing asthma symptoms, but it’s pretty overwhelming to start a new medication with stresses at my work going on. At the same time, the new med could improve my quality of life. I think I can try it.”* Incorporation of an action script can also be transferred to other courses and Colleges of Pharmacy. The final attribute unique to our MI approach is the inclusion of the assessment method of students’ critical reflection upon the impact of the MI exercises. In their own words, students expressed how their eyes were opened to the importance of pharmacist–patient communication and that the choice of words really matters (Table 5). Based on this qualitative assessment, we were able to glean that by participating in the MI skills development exercises, our students experienced growth in professional identity formation “think, act, and feel like a pharmacist” attributes [6] by recognizing the relevance and benefit of honing MI communication skills to future practice. Students expressed that they learned through peer role-plays and found the exercises valuable in improving confidence and thinking “on their feet,” and in increasing their awareness of the nuances and difficulties in communicating empathically with patients.

### 4.2. Impact of Our MI Approach on Student Learning

Our results lend support to the value and feasibility of our MI methods in teaching and learning in pharmacy education. Student reflections indicate that the MI exercise was immersive, placing students in an “under pressure” communication scenario that would be difficult to replicate using conventional pedagogical methods. Moreover, students indicated positive impacts on their intrinsic motivation to improve their skills, knowing that these kinds of real-world counseling activities would likely occur in upcoming IPPE rotations and eventually APPE rotations. The value of immediate feedback was also evident in the experience, as shown in student reflections and our own review of a select number of video recordings and transcripts. Instructing “patients” to engage in change talk only if the “pharmacist” communicated effectively (with empathy and MI principles/techniques) was one specific feature that created a context for this immediate feedback. Finally, this assignment effectively capitalized on the concept of iteration, inviting students to repeat, practice, and refine their communication activities before submitting a final version for a grade. While we do not have systematic evidence of how prevalent repeated practice was in the class cohort, reflections provide anecdotal evidence of multiple attempts, restarts, and re-dos to achieve a stronger performance. In terms of feasibility and cost-effectiveness, the iterative potential of our role-play innovation is another strength. Using professional standardized patients and the resources of a patient simulation lab would likely make multiple attempts and experimentation prohibitively costly. The built-in improvisational aspects of our action script ensured that no playthrough was identical, making this an example of self-sustaining and self-renewing educational material in the hands of students who can work with it at their own pace.

### 4.3. The Benefits of Peer Engagement in Role-Play

Our findings corroborate previous research on the value of using peers, rather than exclusively professional standardized patients (SP), for role-play activities, particularly when the learning is foundational or formative [25,26,27]. Research has shown comparable learning outcomes when using peers or SPs, but peer engagement also brings less tangible yet essential outcomes such as increased trust, confidence, and interconnection within a learning community [10], all of which were evident in our student reflections. As previous research [17] has shown, confidence in MI skills is a strong predictor of the likelihood of implementing them in future practice. In the context of a unit focused extensively on empathic communication, a cornerstone of MI counseling, peer engagement also had several unique benefits. First, during both the formulation challenges written assignment and the MI peer role-play assignment, students had access to a team or a partner to discuss their language formulations. Empathy is ultimately about taking the perspectives of other people; thus, students found value in being able to try out their formulations on other people and hear their reactions. Peers were able to articulate how different word choices struck them connotatively, such as unintentionally domineering or even mildly offensive formulations. Put simply, a good way to understand how other people might interpret your speech is to discuss it with other people. Second, student reflections suggest that the empathic communication they were learning about was extended to their interactions with one another. For example, certain students perceived even the process of scheduling their Zoom conferences as a potential exercise in empathy; as fellow students with active lives in a demanding program, they approached scheduling with understanding and flexibility. Although the development of interprofessional attitudes and teamwork skills was not the centerpiece of this innovation (see Table 1), it is a significant CAPE outcome, and the connection between empathic communication and interprofessional respect cannot be overstated.

### 4.4. Cultivating Communication Skills “On Camera”: Laying Foundations for a Telehealth Future

As this assignment has evolved over the years, and the use of Zoom technology has been adopted at scale in higher education during the COVID-19 pandemic, it has become clear that this technology is not only a convenient modality of communication but also a capable medium to simulate many aspects of telemedicine and to cultivate awareness of communication skills “on camera”. Effective communication through a videoconferencing interface presents unique multi-tasking challenges, such as maintaining consistent eye contact and warm facial expressions through a digital camera, while also managing an electronic medical record (EMR) [32]. While our interface did not include an EMR, it did require participants to maintain a mutual orientation to the unfolding action script in the third video window (screen share), which we believe served as a useful proxy for the kind of multi-tasking that is required for optimal telehealth consultations. These skills will be indispensable to the student pharmacist, as telepharmacy is increasingly adopted to expand healthcare access [33].

### 4.5. Limitations and Future Directions

There are limitations to our work. The thematic analysis of student reflections was limited to a sub-sample of students who self-selected to write about the MI exercises for their final course reflection; we included these data to provide a richer first-hand perspective on the educational merits of this experience. However, we did not ascertain how representative their views were in the full class sample; this study limitation that could be strengthened by conducting student focus groups to gain deeper insights into the teamwork dynamics, perceived importance of the patient role, and the impact of the primer and action script in balancing structure and improvisation in the MI peer role-play exercises. Also, when designing formulation challenges in the Week 2 assignment, students worked in teams of four, and as such we did not assess individual knowledge or competence (as was conducted for the individual peer role-play videos); it is possible that this group dynamic may have masked deficiencies in team members. Further, the peer role-play exercise was introduced and practiced only once before students submitted videos for formal evaluation. Affording more practice opportunities would likely be of benefit to students in communication skills development, not only within a given course [11] but longitudinally across multiple courses within the curriculum [15]. Questions remain about the best modality to ensure that student pharmacists’ learning gains in MI counseling are durable over time [16]; this sets the stage for future research investigating the longitudinal impact of this style of role-play and/or coupling the peer role-play approach with other effective techniques such as the use of a standardized patient [10].

For additional future directions, we plan to share our primer and action scripts concepts with our colleagues in the six-semester Pharmacy Skills course sequence, where MI exercises already occur, in hopes of standardizing the approach. Conceptually, it might make sense to “allow” the use of drug/disease-specific primers and action scripts in P1 and P2, and perhaps minimize their use during P3, when students have a good deal of pharmacotherapeutics and practice with the MI approach under their belt, with a goal of entering P4 without the use of either of these tools. Our new primer and action script innovations are readily transferable to other courses and to other pharmacy colleges.

## 5. Conclusions

This study indicates that well-designed peer role-play can be an effective and meaningful approach to cultivating patient-centered communication skills early in pharmacy curricula. The pedagogical innovations described and evaluated in this study, particularly the fictionalized “Asthma and Medication Primer” and the “Turn-by-Turn Action Script”, provide an engaging and immersive simulation of pharmacy counseling that is well-aligned with P1 students’ levels of clinical experience and pharmacotherapeutic knowledge. In contrast to the use of professional standardized patients, this peer-based approach to role-play is not only resource-efficient but can also build community and connection within a cohort of learners.

## Figures and Tables

**Figure 1 pharmacy-11-00122-f001:**
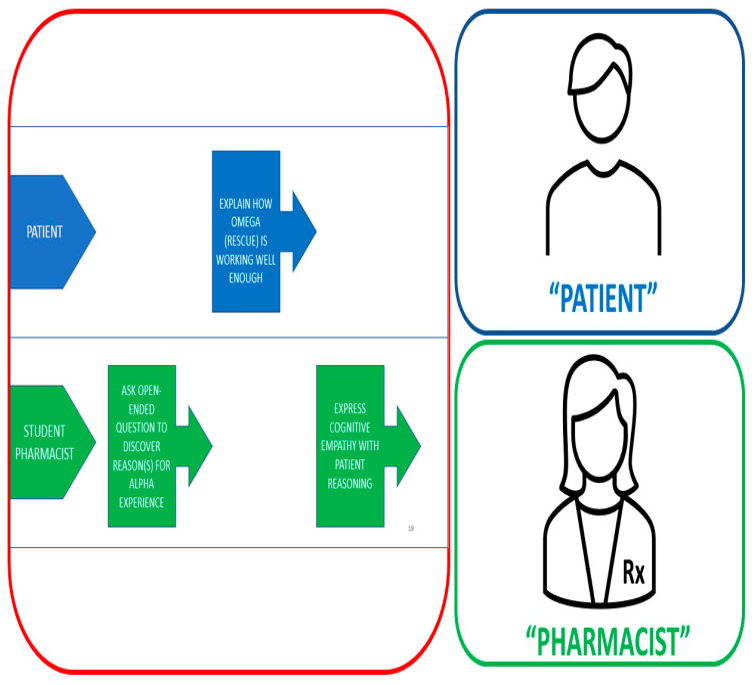
Depiction of Zoom interface with action script (**left**) and role-players (**right**).

**Figure 2 pharmacy-11-00122-f002:**
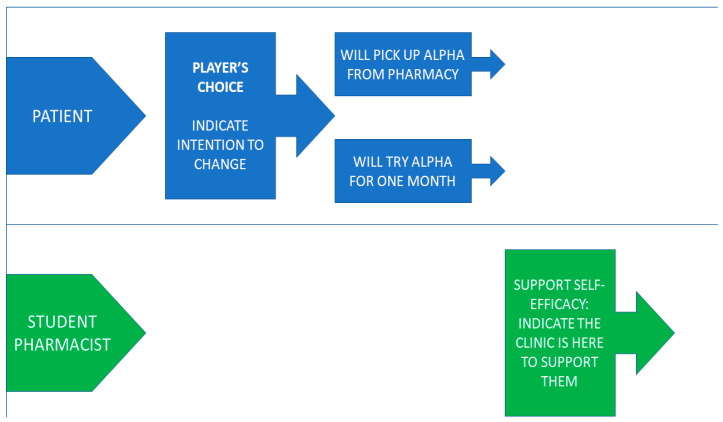
Example of player’s choice in the turn-by-turn action script.

**Table 1 pharmacy-11-00122-t001:** Ten educational CAPE outcomes addressed through MI peer role-play and formulation challenges learning objectives.

MI Learning Objectives after Completing MI Exercises, Students Should Be Able to:	CAPE Outcome
Affirm the importance of empathic communication between patients and pharmacists.	3.6, 4.1
Demonstrate reflective listening with peers who are role-playing as patients.	3.6
List the steps necessary to decrease issues or relational resistance during counselling.	1.1
Discuss the connections between rapport (relationship) building, reflective listening, and issue/relational resistance.	1.1
Demonstrate reflective listening as you paraphrase the patient’s sense making.	3.2, 3.3, 3.6
Collect the patient’s sense making as neutral, non-judged data.	3.3, 3.6, 4.4
Invite patients to reflect on their own motivations for change.	3.2, 3.3, 3.6
Close the deal with goal setting and conditional commitment with high rapport.	2.3, 3.3, 3.6
Through peer interactions, demonstrate the ability to verbally communicate clearly and professionally, working toward shared goals.	3.4, 3.6, 4.2, 4.4
Through engagement in innovative role-playing exercises with peers, gain experience in patient–provider communication and patient advocacy.	3.2, 3.3, 3.6, 4.3

MI = Motivational interviewing; CAPE = Center for the Advancement of Pharmacy Education [28]; 1.1 = learner; 2.3 = health and wellness promoter; 3.2 = educator; 3.3 = patient advocate; 3.4 = interprofessional collaborator; 3.6 = communicator; 4.1 = self-aware; 4.2 = leader; 4.3 = innovator and entrepreneur; 4.4 = professional.

**Table 2 pharmacy-11-00122-t002:** Instructions provided to peer role-players for playing pharmacist and patient roles.

Student Pharmacist Role	Patient Role
Your patient is currently relying only on Omega to manage their asthma symptoms when they become severe. You and your patient’s physician both agree that your patient would experience better health by taking Alpha every day. Your goal is to increase your patient’s intrinsic motivation and self-efficacy to adhere to an Alpha regimen.	You were recently seen in the emergency room for a severe asthma attack. Recent changes in your life have worsened your asthma and you have significant stress that has made it difficult for you to adhere to your physician’s recommended asthma action plan. Your goal is to resist taking Alpha (because you prefer Omega) for much of the encounter but to engage in reasonable change talk by the end if the student pharmacist communicates with you effectively.

**Table 3 pharmacy-11-00122-t003:** Peer role-play motivational interviewing criteria where competency was not fully met.

Rubric Criteria:	# Students (out of 82)
Employs patient autonomy	19
Expresses empathy	1
Develops discrepancy	23
Enhances patient’s self-efficacy	3
Rolls with resistance	6
Patient role: well-prepared	9
Patient role: challenge presented	10

**Table 4 pharmacy-11-00122-t004:** Examples of quotes from student role-playing videos demonstrating motivational interviewing ^1^ principles and techniques.

Role Played	MI Rubric Criteria	Context from Action Script	Illustrative Quote Demonstrating Need of Improvement to Meet Competency in MI Skill	Illustrative Quote Demonstrating Competency in MI Skill
Student Pharmacist	Employs patient autonomy: reassures patient that they are in control	Patient expresses that Omega (rescue) inhaler has been working well, so why take new medication (Alpha for daily maintenance)?	“But I want you to take the Alpha daily…”	“I want to show you that you have options, I want to educate you on the options you have. It is totally your choice…”
Student Pharmacist	Expresses empathy: summarizes ambivalence	Patient explains to the pharmacist why they do not take the Alpha as the MD recommended.	“I completely understand that it is very hard to remember taking medications. I take medications myself and I sometimes forget to take my medications.”	“I get that. On the one hand, you have to work and worry about the stresses at work. And on the other hand, you have to worry about, you know, taking another medication along with you every day.”
Student Pharmacist	Develops discrepancy: provides patient opportunity to talk about distance between current and goal state	Patient is explaining why they have not adhered to Alpha in the past.	“Is there any way you would be more willing to try the Alpha, or do you feel as though it is not necessary at all?”	“How would you feel if your asthma meds worked better, and you had an easier time breathing?”
Student Pharmacist	Enhances patient’s self-efficacy: helps patient build confidence through change-talk	Patient is trying to determine if they are up to the task of taking on a new maintenance medication (Alpha inhaler), given stresses in their life and possibly forgetting to take the med every day.	“I recommend you take it. It is the best course of action for you.”	“We are here to help you. Would you consider a medication alert using your phone to remind you to take the daily medication?”
Student Pharmacist	Rolls with resistance: non-confrontational, shifts topic, stresses patient autonomy	Pharmacist is asking the patient how they take the Alpha. Patients are responding that they do not take it at all or as prescribed.	“If you only take Alpha like once in a while, it won’t help you by preventing your symptoms.”	“I get that you don’t want people (family) nagging you (to take maintenance Alpha). Really, the entirety of this (counseling session) is centered on you, what you are willing to do (to improve your asthma treatment). That said, is there anything you would be willing to do with the Alpha medication at this time?”

^1^ Motivational interviewing = MI.

**Table 5 pharmacy-11-00122-t005:** Themes derived from analysis of student reflections.

Theme	N (%) *	Explanation	Sample Quotation
Relevance to current and future professional identity formation	54 (46)	Students perceived that this experience would be beneficial in their short- and long-term futures, including upcoming IPPE rotations, pharmacy skills labs, APPE rotations, and professional practice. They also indicated the breadth of applicability beyond the clinical realm, including leadership, teamwork, community service, and interpersonal relationships.	“Communication is a huge part of my transformation into a practicing professional. I will utilize these skills and behaviors in all parts of my life such as interviews, peer-to-peer communication, relationships, parenting, managing, work, and public service. This class has provided endless utility to me and communicating in a way that is professional is just the tip of the iceberg.”
Communication skill awareness and development	51 (44)	Students experienced “aha” moments regarding the importance of word choice. They described the challenges of using open-ended questions to guide patients, rather than expert recommendations. They appreciated the impact of subtle variations in empathic responses, such as the difference between “I understand” and “That’s understandable.” They also indicated that communicating through digital media added value to the experience.	“I work as a pharmacy technician, so I talk to many patients every shift I work, but while counseling I really had to think about how I was saying everything. I had never really thought about how I was saying things to patients, but it really does make a difference. The motivational interviewing assignment changed the process of what goes through my head while talking to patients and that will stick with me for many years, through school and long after I become a pharmacist.”
Learning MI principles and techniques	36 (31)	Students indicated awareness of the four main MI techniques and underlying principles, with “expressing empathy” emerging as the most consistently mentioned. Many articulated difficult balances and trade-offs among the principles and techniques, such as respecting autonomy and avoiding confrontation while still guiding patients to make healthier choices.	“When I first tried to do the video without thinking about what I wanted to say, I realized how hard it is to have a conversation and motivate a patient that is challenging you because you still have to help direct them to make the best decision for their health, while also making them feel understood. Learning how to roll with resistance without being confrontational is very difficult, and I think that’s one of the main things that I learned from this assignment.”
Value of peer engagement	31 (26)	Students indicated that working with peers helped build connection, trust, and confidence. The experience created a dynamic of interdependence and mutual accountability. Many indicated that discussing empathic responses with peers was instructive, as it helped them appreciate the different ways that others might interpret their empathic formulations, including unintended meanings.	“We began to record [my partner’s] part of the assignment, and once we finished recording, I immediately apologized to her because I thought that I messed up her part, but surprisingly she told me that it was perfect and that I did amazingly. We continued for my part of the assignment, and I started to mess up a lot and immediately became self-conscious of how badly I was doing, but my partner assured me that I was doing my best and to never give up. I don’t know why, but at that moment I felt more confident and remembered that it’s fine to mess up, so I gave it my all in my part of the video, and once we finished, my partner said how proud she was of me.”
Learning through role-play	20 (17)	Students indicated that the action script helped to surface realistic challenges they would encounter in professional practice, in some cases exposing areas for improvement in their communication skills/approach. Many indicated that a “patient” role-player had helped draw attention to these deficits through their responses as the “patient.” Several students mentioned the pressure of improvising on the spot but nonetheless perceived the purpose and value of the pressure for their development.	“The [action script] included challenges such as patient reluctance and noncompliance, which I thought were extremely helpful in simulating a real pharmacist–patient interaction. Not having a full script made the scenarios more realistic and I had to think under pressure while the patient was waiting for a response. This “under pressure” feeling cannot be simulated without working with peers or in a group… I felt more comfortable in my abilities to overcome obstacles that may arise during pharmacist–patient counseling after this activity due to the realistic parameters of the assignment.”

* Because a single text extract can be tagged with multiple codes, the extract counts shown here exceed the total of 117 extracts; the percentage refers to the number of extracts out of 117 that contained material relevant to the theme.

## Data Availability

Data sharing is not applicable to this article.

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
