# Peer review of "Exploring the Impact of an Innovative Peer Role-Play Simulation to Cultivate Student Pharmacists’ Motivational Interviewing Skills"

_pharmacy, 2023, doi:10.3390/pharmacy11040122_

Round 1
Reviewer 1 Report
This was an extremely well written paper. This topic is important and timely and the methods provided a solid roadmap for faculty looking to implement a similar activity. I found the description comprehensive and easy to follow. It was long, but I didn't identify a specific section that could be reduced and maintain the clarity, so I think it is ok as long as it is within the correct word count.
I only have one suggestion:
Introduction, line 35: The sentence "The pharmacy profession...tobacco cessation" is very long - would suggest breaking it into two sentences.
Author Response
I only have one suggestion:
Introduction, line 35: The sentence "The pharmacy profession...tobacco cessation" is very long - would suggest breaking it into two sentences.
Response: We have made this sentence shorter, per the reviewer’s suggestion, as follows:
The pharmacy profession has a long history of counseling patients about safe and effective medication use; many patients are struggling to afford and/or adhere to complex medication regimens for chronic illnesses and behavioral health challenges like tobacco cessation [3].
The change is shown in “track changes” on the manuscript, attached below.

Author Response
Reviewer #2: Comments to the Authors
Thank you for the opportunity to review this article. This article describes an innovative approach to teaching junior pharmacy students communication skills. It will be of interest to pharmacy educators who teach pharmacy skills courses. The article is well written and the details of the intervention of clearly described. The authors should be commended for the willingness to disseminate their teaching materials within the manuscript and were generous in their inclusion of activities, rubrics, and teaching resources in the supplementary appendices.
I feel there are opportunities to shorten and provide greater focus to certain sections of the manuscript, particularly section 1.2 of the introduction and section 2.1, and 2.3 of the methods. Response: We have extensively shortened the manuscript per the reviewer’s suggestions, explained in sections below.
Specific comments below:
Abstract: The abstract clearly explains background, purpose, methods (mixed method), results and conclusions.
Introduction: The introduction provided excellent, though detailed, background and stated the purpose of the study.
Overall, I feel the introduction is tackling “too much”. The authors provide detailed reviews of the PC process, MI, gamification and merits and pitfalls of role-playing vs. SPs. To enhance readability, I would recommend clarifying the problem you are tackling: for example, the shortcomings of peer-role play. With the problem more clearly stated the significance your interventions to iterate and simulations involving peer role-playing for students can be more effectively emphasized. Response: we concur that perhaps we were tackling too much, and have removed gamification form the manuscript, including all citations. This is noted in ‘track changes’ throughout the manuscript.
Examples of sections that could be condensed:
Section 1.2: gamification – the link to study purpose is not as strong. There is opportunity to streamline 79-98. Response: this section 2.1 streamlined to remove gamification. Reference 20 may not be the best reference choice for that statement. Response: Reference 20 removed, and three different references from pharmacy educators inserted (Biddle, Kane-Gill (new to manuscript), Zerilli (new to manuscript), as shown in ‘track changes’ in the manuscript.
I would recommend looking for a more relevant reference for line 98-99 as the point being made is impactful however the reference (26) is not situated within a healthcare context. Response: replaced reference 26 with pharmacy educator reference (Wallman), shown in track changes in manuscript.
Methods:
Overall, the methods are clear, and well written. While the description of the evolution of the role-playing assignment was informative, in my view, not all the detail presented is required, and the methods may benefit from being shortened. Response: we have considerably shortened the Methods section, per reviewer’s suggestions, as explained below.
For example, Section 2.1, lines 142-180 could be condensed. It’s unclear the significance that the detailed description of the preparatory activities adds to the manuscript. I feel these could be summarized, and samples (e.g. Table 1) included in the appendices. Response: Section 2.1 is condensed per suggestion, and Table 1 is now placed in supplemental materials. All remaining tables in the text have been renumbered. This is shown in track changes.
Section 2.2.1 repeats some information from the introduction. I feel this is best accomplished in the introduction as a “problematization of the issue”. Response: Section 2.2.1 is now deleted; see track changes.
Sections 2.3 were well done – though some of the narratives on how challenges were overcome were expansive and could be shortened (e.g. lines 319-355) Response: text is condensed per reviewer’s suggestion. See track changes in the manuscript.
Section 2.4 – Lines 380-382 (the example presented) could be removed. Response: example is removed from text, see tack changes in manuscript.
The qualitative methods described for the analysis of the reflections are robust.
Results:
Table 4 was difficult to understand in relation to average grade (of 9.4/10 – which seems high). Consider adding an explanatory sentence. Response: An explanatory sentence has been added, per suggestion. See track changes in manuscript.
A grade distribution between top 10 and bottom 10 might be interesting considering the overall high average. Response: Perhaps, though the more elucidating component of our assessment of this MI exercise was the qualitative analysis of the reflections, as shown in Tables 5 and 6 (now renumbered Tables 4 and 5) and discussed in Discussion. No changes were made to the manuscript.
Table 5 and 6 highlighted the student performance and added to the readers' understanding. Quotes were well chosen and were representative.
Discussion The discussion was well done with relevant literature referenced, discussion of the education tool and how it impacted students as well as future plans to enhance the tool and the curriculum.
Conclusion: While the journal does not require a formal conclusion, given the complexity and evolution of this activity, a conclusion may be helpful in articulating the key findings for readers. Response: A conclusion has been added, per suggestion.

Reviewer 3 Report
Thank you for this detailed report of your innovative P1 course. I would suggest the authors consider submitting much of the background in this report as a separate review article. It is extremely long for a research report and the reader cannot follow the connection to the report. The authors refer to this work as a study, however, I do not consider the methods used as research. The authors provide in the results student evaluations and open-ended responses from their course evaluations. This doesn't appear to be research to me.
Similarly, the methods are extremely long - almost 10 pages. I cannot read through 10 pages of methods and clearly understand what was done. Much of what is in the methods describes why the authors designed aspects of the course as they did. There is too much detail here for a research study.
Other comments:
-On page 13 (line 397), the information on the number of students submitting reflections should be in the results section.
- Results (page 14). I think the results from Table 4 could be explained better. For example, did only 1 out of 82 students express empathy during the video?
-Results - Table 5. I don't think examples of quotes demonstrating competency in MI is necessary. This is a lot of detail and most of us in pharmacy education know what MI competency looks like.
- Discussion (Page 19 Section 4.2). I understand that the authors used concepts of gamification in this exercise... but I wouldn't consider this exercise anything like gamification. So I feel talking about the MI approach in the concept of gamification is somewhat misleading.
There are a few instances of extremely long sentences (over 5 lines in length) that should be broken up so that the reader can follow the concept. There are also a few instances of phrases modifying the wrong words.
Author Response
Reviewer #3:
Thank you for this detailed report of your innovative P1 course. I would suggest the authors consider submitting much of the background in this report as a separate review article. It is extremely long for a research report and the reader cannot follow the connection to the report. Response: the introduction section has been shortened by eliminating the gamification sections. The remaining sections on MI, simulated role-play set the foundation for the MI intervention, and thus were not removed. See track changes in the manuscript.
The authors refer to this work as a study, however, I do not consider the methods used as research. The authors provide in the results student evaluations and open-ended responses from their course evaluations. This doesn't appear to be research to me. Response: This is a mixed-methods study that included analysis of two forms of empirical data: 1) numerical scores on an instructor-graded assignment rubric and 2) textual commentary produced by students in their final written course reflection assignment. We may be misunderstanding the reference to "student evaluations" and "course evaluations," but we wanted to clarify the sources of data that were analyzed as evidence of student learning and impact. We are aware that there are different research paradigms, including quantitative and qualitative approaches, so we are unsure of the criteria or paradigm from which this manuscript wouldn't be considered research. While our approach is not variable-analytic or hypothesis-driven, it is consistent with the theory-generating aims of analytic induction common in qualitative research. If additional specific concerns about the methods (or the relationship between the methods and resulting claims) can be expressed by the reviewer or editor, we would be happy to respond.
Similarly, the methods are extremely long - almost 10 pages. I cannot read through 10 pages of methods and clearly understand what was done. Much of what is in the methods describes why the authors designed aspects of the course as they did. There is too much detail here for a research study. Response: the Methods section has been considerably condensed per the reviewers’ suggestions, including the removal of Table 1. See track changes in the manuscript.
Other comments:
-On page 13 (line 397), the information on the number of students submitting reflections should be in the results section. Response: done, see manuscript track changes.
- Results (page 14). I think the results from Table 4 could be explained better. For example, did only 1 out of 82 students express empathy during the video? Response: an explanatory sentence has been added to Table 4 (now renumbered Table 3). See track changes.
-Results - Table 5. I don't think examples of quotes demonstrating competency in MI is necessary. This is a lot of detail and most of us in pharmacy education know what MI competency looks like. Response: Table 5 (now renumbered Table 4), was included to give readers a qualitative sense of the actual MI intervention. Reviewer 1 had no issue with this Table, and Reviewer 2 commented “Table 5 and 6 highlighted the student performance and added to the readers' understanding. Quotes were well chosen and were representative.” Thus, we have elected to keep this Table in the manuscript.
- Discussion (Page 19 Section 4.2). I understand that the authors used concepts of gamification in this exercise... but I wouldn't consider this exercise anything like gamification. So I feel talking about the MI approach in the concept of gamification is somewhat misleading. Response: Gamification has been removed from the manuscript. See track changes.
Comments on the Quality of English Language
There are a few instances of extremely long sentences (over 5 lines in length) that should be broken up so that the reader can follow the concept. Response: Sentence in Introduction that was pointed out by Reviewer #1 was shortened. A few other long sentences were eliminated in the many paragraphs that were deleted. There are also a few instances of phrases modifying the wrong words. Response: we are unsure which sentences this comment pertains to, and refer to editors.
